# Synergistic Effect of Surface Acidity and PtO*_x_* Catalyst on the Sensitivity of Nanosized Metal–Oxide Semiconductors to Benzene

**DOI:** 10.3390/s22176520

**Published:** 2022-08-29

**Authors:** Artem Marikutsa, Nikolay Khmelevsky, Marina Rumyantseva

**Affiliations:** 1Chemistry Department, Moscow State University, Vorobyovy Gory 1-3, 119234 Moscow, Russia; 2LISM, Moscow State Technological University Stankin, Vadkovsky Ln. 1, 127055 Moscow, Russia

**Keywords:** semiconductor metal oxides, gas sensor, benzene, active sites, platinum, supported noble metal catalyst

## Abstract

Benzene is a potentially carcinogenic volatile organic compound (VOC) and its vapor must be strictly monitored in air. Metal–oxide semiconductors (MOS) functionalized by catalytic noble metals are promising materials for sensing VOC, but basic understanding of the relationships of materials composition and sensors behavior should be improved. In this work, the sensitivity to benzene was comparatively studied for nanocrystalline *n*-type MOS (ZnO, In_2_O_3_, SnO_2_, TiO_2_, and WO_3_) in pristine form and modified by catalytic PtO*_x_* nanoparticles. Active sites of materials were analyzed by X-ray photoelectron spectroscopy (XPS) and temperature-programmed techniques using probe molecules. The sensing mechanism was studied by in situ diffuse-reflectance infrared (DRIFT) spectroscopy. Distinct trends were observed in the sensitivity to benzene for pristine MOS and nanocomposites MOS/PtO*_x_*. The higher sensitivity of pristine SnO_2_, TiO_2_, and WO_3_ was observed. This was attributed to higher total concentrations of oxidation sites and acid sites favoring target molecules’ adsorption and redox conversion at the surface of MOS. The sensitivity of PtO*_x_*−modified sensors increased with the surface acidity of MOS and were superior for WO_3_/PtO*_x_*. It was deduced that this was due to stabilization of reduced Pt sites which catalyze deep oxidation of benzene molecules to carbonyl species.

## 1. Introduction

The search for sensitive and low-cost detectors of volatile organic compounds (VOC) is motivated by the needs of environmental safety, healthcare, non-invasive medical diagnostics, etc. Metal–oxide semiconductor (MOS) sensors are efficient for detecting VOC at the ppb-ppm concentration level in air [1,2]. Extensive research has been carried out to improve the sensitivity and selectivity of conventional sensors based on zinc oxide, indium oxide, tin oxide, tungsten oxide, and titania [3,4,5]. The search for sensors with improved behavior is mainly focused on the design of new morphologies of nanomaterials, elaboration of new mixed-metal oxides, and composites with catalytic additives (noble metals and transition-metal oxides), heterojunctions (*p-n*, *n-n*, and *p-p*), and graphene-based materials [3,5,6,7,8,9]. However, there is a lack of basic understanding of the relationships between the fundamental properties of sensor materials and sensing behavior in the detection of VOC. Benzene (C_6_H_6_) is a potentially carcinogenic pollutant emitted into the air from oil and coal industries, plastics, solvents, and smoke. The threshold limit value (8−h TWA) for benzene recommended by OSHA is 1 ppm [10]. Benzene is a primary aromatic compound. Establishing the trends in sensitivity of noble-metal-loaded MOS may open a way towards rational design of sensors with improved sensitivity to aromatic VOC. Extensive research of materials for sensing benzene and other aromatics (toluene and xylene) have been performed. It was mainly focused on *n*-type and *p*-type MOS with various morphologies (nanowires, nanoparticles, and hollow spheres) as heterojunctions and nanocomposites with noble metal nanoparticles [11,12,13,14,15,16]. However, in the literature there is a lack of systematic comparative studies of materials that have different chemical composition but similar morphologies, which would be helpful in establishing the fundamental regularities in sensing benzene.

Key processes of gas sensing consist of target molecules’ adsorption and redox conversion by chemisorbed or lattice oxygen at the oxide surface. It is similar to the functioning of heterogeneous catalysts. There are a number of fundamental approaches for the design of VOC oxidation catalysts based on metal oxides and supported noble metals (Pt, Pd, and Au) that rely on energetic and structure–activity relations [17,18,19,20]. Conventionally, metal–oxygen bond energy (*E*_M-O_) has been considered as a descriptor of surface oxygen activity in the catalyzed reactions [21,22,23]. It is challenging to distribute such an approach in the field of gas sensors. In our previous work, we showed that the trends in sensitivity of Au-modified MOS to acetone and methanol could be rationalized in terms of surface oxygen reactivity which was tailored by optimal metal–oxygen bond energy in metal oxides [24].

This work is focused on a comparative study of active surface sites and sensitivity to benzene of nanocrystalline metal oxide semiconductors-ZnO, In_2_O_3_, SnO_2_, TiO_2_, and WO_3_-in pristine form and modified by catalytic PtO*_x_* nanoparticles. The aim of the present work was to analyze the correlations between the sensitivity to benzene and fundamental parameters of sensing materials. It is challenging to determine the possible reasons for the differences in materials’ sensitivity to benzene considering the impacts of active sites at the materials surfaces— acid sites and oxidation sites—which were probed by temperature-programmed techniques. Distinct trends were observed in the sensitivity of pristine and PtO*_x_*−modified MOS using metal–oxygen bond energy as a descriptive parameter. These correlations were rationalized by the impacts of active sites at the materials’ surfaces and PtO*_x_* nanoparticles in the sensing mechanism, which was studied by in situ infrared (DRIFT) spectroscopy.

## 2. Materials and Methods

The materials investigated were nanocrystalline *n*-type MOS—ZnO, In_2_O_3_, SnO_2_, TiO_2_, and WO_3_—which have been conventionally used in gas sensors. The oxides possess wide bandgap energies in the range 2.5–3.6 eV and distinct fundamental parameters: cationic charge and radius, metal–oxygen bond energy, and formation enthalpy (Table 1).

### 2.1. Material Preparation

The materials were obtained by chemical aqueous deposition techniques previously described in detail [24,29]. ZnO, In_2_O_3_, SnO_2_, and TiO_2_ were synthesized by deposition of metal hydroxides from aqueous solutions with subsequent heat treatment. Analytical pure grade Zn (CH_3_COO)_2_·2H_2_O, InCl_3_·4H_2_O, SnCl_4_·5H_2_O, and TiCl_4_ (Sigma-Aldrich, St. Louis, MO, USA) were used as precursors. An excess of 1 M aqueous ammonia was added dropwise to the stirred solutions of precursors. The final pH = 6 was reached for the deposition of SnO_2_∙nH_2_O and TiO_2_∙nH_2_O, and Zn (OH)_2_ and In (OH)_3_ were deposited at pH = 9. Tungstic acid was deposited at 80 °C by the addition of 7.8 M nitric acid to 16 mM ammonium paratungstate (APT) solution in the volumetric ratio 2.6:1. The APT solution was prepared from (NH_4_)_10_W_12_O_41_·5H_2_O (Sigma-Aldrich, >99%). The deposits were washed by deionized water, centrifuged, and dried at 80 °C. Nanocrystalline ZnO, In_2_O_3_, SnO_2_, and WO_3_ were obtained after the calcination of metal hydroxides at 300 °C for 24 h. TiO_2_ was annealed at 700 °C for 24 h. Nanocomposites MOS/PtO*_x_* (1 wt.% Pt) were synthesized by impregnation of metal–oxides powders with ethanol solution of Pt(acac)_2_ (Sigma-Aldrich, 98%). The obtained suspensions of MOS/Pt(acac)_2_ were dried and annealed at 220 °C for 24 h to decompose platinum (II) acetylacetonate.

### 2.2. Material Characterization

X-ray powder diffraction was measured using a DRON-3M diffractometer, Cu Kα radiation (λ = 1.5406 Å). Crystallite size was calculated by the Sherrer equation using full width of half maximum of most intense peaks. Specific surface area (S_BET_) was measured by nitrogen adsorption by the Brunauer-Emmett-Teller (BET) method in the single-point mode (*p*/*p*^0^ = 0.3) using the Chemisorb 2750 (Micromeritics, Norcross, GA, USA) instrument equipped with thermal conductivity detector. X-ray photoelectron spectroscopy (XPS) was performed on the XPS system (Thermo Fisher Scientific, Waltham, MA, USA) equipped with a hemispherical analyzer using monochromatic Al Kα radiation (1486.7 eV); binding energy (BE) was calibrated by C 1s peak at 285.0 eV. Transmission electron microscopy (TEM) and high-resolution TEM (HRTEM), selected area electron diffraction (SAED), and scanning transmission electron microscopy in high-angle annular dark-field mode (STEM-HAADF) were performed using a Libra 200 microscope (Carl Zeiss) with an accelerating voltage 200 kV. The energy-dispersive X-ray spectroscopy (EDX) signal was recorded on a silicon drift X-MAX 80 T detector.

Temperature-programmed reduction (TPR) by hydrogen was registered using the Chemisorb 2750 (Micromeritics) instrument. The samples (20 mg) were pretreated in dry air at 200 °C to desorb humidity and cooled down to room temperature. TPR was registered during the samples’ heating to 900 °C at the rate of 10 °C/min under the flow of H_2_ (8 vol.%) in Ar with the flowrate 50 mL/min. Diffuse-reflectance infrared Fourier-transformed (DRIFT) spectra were registered by Frontier (Perkin Elmer, Waltham, MA, USA) spectrometer at ambient conditions with automatic H_2_O/CO_2_ compensation. DiffusIR annex and heated flow chamber HC900 (Pike Technologies, Fitchburg, WI, USA) sealed by KBr window were used for in situ spectra registration in the wavenumber range 4000–700 cm^−1^ with the resolution 4 cm^−1^, averaging 30 scans. Powders (50 mg) were placed in alumina crucibles (6 mm diameter) and pretreated in dry air at 150 °C. DRIFT spectra were registered at room temperature under the gas flow (100 mL/min) of NH_3_ (200 ppm) and at temperature 25–220 °C under the flow of benzene (10 ppm) in air with the reference to pure air.

Temperature-programmed desorption of ammonia (TPD) was measured by the Chemisorb 2750 (Micromeritics) instrument. Quadrupole mass spectrometer was used to analyze the desorbed gas. Samples (100 mg) were granulated and the fraction with 0.25–0.50 mm-sized grains was tested. After pretreatment in He at 200 °C for 1 h and in dry air for 1 h, the samples were cooled to room temperature and held under the flow of NH_3_ (8 vol.%): He gas mixture. Then, the samples were evacuated in He at 50 °C for 20 min to desorb the physically adsorbed ammonia. TPD patterns were registered in the flow of He (30 mL/min) while heating the test tube with samples to 800 °C at the rate of 10 °C/min. Mass spectra of desorbed gas were acquired for mass numbers 16, 17, 18, 28, 30, 32, and 44.

### 2.3. Evaluation of Sensitivity of Sensor

Sensors were prepared as thick films (~10 μm) deposited on alumina substrates embedded in TO-8 packages. The sensor substrates were provided vapor-deposited Pt contacts (size 0.3 × 0.2 mm, gap 0.2 mm) and Pt meander. Sample powders were dispersed in terpineol and drop-deposited by a micropipette. The images of substrates before and after thick film deposition are in Supplementary data (Appendix A). The sensors were tested by a PC-controlled electrometer with a gas flow chamber. Sensors were heated at 220 °C for 12 h to remove organic binder and form the sensing layer. DC-resistance was measured with the applied voltage 1.3 V at temperature 100–220 °C. The reference gas was pure air from a generator of pure air model “2.0–3.5” (Himelectronica, Moscow, Russia); contamination level was in the limit of 10 ppm H_2_O, 2 ppm CO_2_, 0.1 ppm hydrocarbons. Certified gas mixture C_6_H_6_:N_2_ (52 ± 4 ppm, Monitoring) was used as the source of target gas. Gas flows were controlled by mass-flow controllers EL-FLOW (Bronkhorst, Ruurlo, The Netherlands). Pure air and C_6_H_6_:N_2_ flows were mixed using pipelines and fittings (Camozzi, Brescia, Italy) to prepare the test gas with different benzene concentrations. Gas flowrate was 100.0 mL/min in the sensing tests. Sensor signal *S* was defined as relative resistance change according to Equation (1):*S* = (*R_a_* − *R_g_*)/*R_g_*,(1)
where *R_a_* is resistance in air, and *R_g_* is resistance in test gas.

## 3. Results

### 3.1. Material Characterization

The compositions and parameters of microstructure of materials—crystallite size (specific surface area evaluated by XRD) and BET measurements—are summarized in Table 2. The phase compositions of pristine metal oxides were similar to those of *n*-type MOS in the previous work [24]. Single-phase wurtzite-like ZnO, cubic In_2_O_3_, rutile-like SnO_2_, and monoclinic γ-WO_3_ were obtained according to XRD (patterns are in Supplementary data, Appendix A). Mixed-phase composition of TiO_2_ was the result of incomplete transition of anatase to the more thermodynamically stable rutile during the calcination at 700 °C.

In the PtO*_x_*−modified samples, no crystalline phases related to platinum were detected by diffraction tools. Investigation of samples’ microstructure by electron microscopy revealed a mostly irregular spherical shape of metal–oxide nanoparticles (Figure 1). The majority of MOS nanoparticles had the size that agreed with the crystallite size estimated by XRD within approximately *d*_XRD_ ± 5 nm intervals (Table 2). The nanoparticles were agglomerated into aggregates with the size of 30–250 nm. Electron diffraction patterns were relevant to MOS phases detected by XRD; no contribution from platinum was observed. However, Pt-enriched nanoparticles segregated at the surface of agglomerated MOS nanoparticles were revealed by HAADF-STEM and EDX mapping (Figure 1e,f). The particle size distribution of PtO*_x_* was in the range 3–15 nm with the majority of nanoparticles having an average size 5–10 nm, independent of the supporting MOS.

XPS analysis confirmed the purity of obtained materials and the elements’ oxidation states were determined. In the spectra of pristine MOS, only the peaks of oxygen and corresponding metals in the relevant oxidation states were observed: Zn^+2^ in ZnO, In^+3^ in In_2_O_3_, Sn^+4^ in SnO_2_, and Ti^+4^ in TiO_2_ (Figure 2a). The Tungsten W 4f signal was asymmetric and could be deconvoluted into two doublets. The major signal was due to W^+6^ oxidation state. The minor doublet was due to W^+5^ cations. The percentage of W^5+^ in pristine WO_3_ was approximately 10 at. % of the overall W amount. The presence of reduced W^5+^ cations indicated a significant oxygen deficiency in WO_3-δ_ (δ ≈ 0.05) under UHV conditions of the XPS measurements. Oxygen O 1s signals were dominated by the peak at 530.0 ± 0.5 eV due to O^2−^ anions in the oxides bulk (Figure 2b). The minor O 1s peak due to surface species (surface anions, OH-groups, and adsorbed oxygen) was distinguished at higher BE = 531.0–532.5 eV. The higher ratios of surface-to-bulk O 1s peaks areas for In_2_O_3_ and SnO_2_ correlated with the larger specific surface areas of these oxides (Table 1).

The introduction of PtO*_x_* did not affect the oxidation states of metal cations and oxygen anions, but the positions of Ti^4+^, W^6+^, and O^2−^ peaks shifted to higher binding energies. This may be due to a catalyst-supported electronic interaction, namely, electron donation from TiO_2_ and WO_3_ to the supported PtO*_x_* nanoparticles. Platinum was observed mainly in the Pt^2+^ oxidation state with the Pt 4f_7/2_ peak position at 72.5–74.0 eV (Figure 2c). Two Pt^2+^ states were observed in the ZnO—supported sample; the one with the Pt 4f_7/2_ signal centered at 72.8 ± 0.2 eV is indicative of hydrated Pt(II) oxide because it is close to the database value BE = 72.6 eV for Pt (OH)_2_. It was the predominant state of platinum in all the MOS/PtO*_x_* nanocomposites. The other state with the peak Pt 4f_7/2_ at 74.0 eV relevant to PtO (the database value BE = 74.2 eV) was observed only in the spectrum of ZnO/PtO*_x_*. In the spectra of TiO_2_- and WO_3_-based nanocomposites, the peaks of reduced Pt^0^ state were observed at 71.5 ± 0.2 eV. From the peaks area ratio, the fraction of Pt^0^ was estimated to be 27–28 at.% of the total Pt content in TiO_2_/PtO*_x_* and WO_3_/PtO*_x_*.

### 3.2. Active Sites at the Surface of Materials

TPR patterns are shown in Figure 3. Hydrogen consumption bands of In_2_O_3_, SnO_2_, and WO_3_ at temperatures above 350–600 °C were due to reduction of metal–oxide bulks to metals. Reduction of WO_3_ was completed at a higher temperature (600–950 °C) relative to SnO_2_ and In_2_O_3_ (350–650 °C). This agrees with the larger stability of tungsten oxide: higher metal–oxygen bond energy and lower enthalpy of formation, in comparison with other MOS (Table 1). The weak H_2_ consumption bands below 300–350 °C were attributed to the reduction of surface species (oxidative sites), such as metal cations (i.e., oxygen vacancy formation), chemisorbed oxygen, and hydroxyl groups. Zinc oxide and titania could not be reduced completely in the TPR experiments and demonstrated continuous H_2_ consumption at temperatures above 80–200 °C. To uniformly compare the concentrations of oxidative sites at the surface of different samples, the equivalent H_2_ amounts consumed below the conditional threshold temperature 300 °C were quantified. The results are shown in Figure 4a.

In the PtO*_x_*−modified materials, reduction of surface species and bulk oxides (SnO_2_, In_2_O_3_, and WO_3_) started at a temperature that was lower by 50–100 °C with respect to pristine MOS (Figure 3). This indicates the catalytic effect of Pt on the reactions involving hydrogen. In the calculation of oxidative sites concentration at the surface of nanocomposites (Figure 4a), the uptake of hydrogen was taken into account for the reduction of certain amounts of Pt^2+^ determined by XPS.

Figure 4b demonstrates the concentrations of acid sites at the surface of MOS evaluated by temperature-programmed desorption (TPD) of ammonia. The obtained results were almost the same as those described previously [24]. The data for ZnO and In_2_O_3_ were revised in this work, because the oxides were prepared differently than in the cited work. Broensted-type (acidic OH−groups) and Lewis-type acid sites (coordinately unsaturated surface cations) were determined by quantifying ammonia desorption at temperatures 50–200 °C and 200–600 °C, respectively [30]. TPD patterns and mass spectra of evolved gases are in Supplementary data (Appendix A). The acidity of nanocomposites MOS/PtO*_x_* could not be determined by TPD because of PtO*_x_*-catalyzed ammonia oxidation to N_2_O, NO, and H_2_O prior to NH_3_ desorption, as was detected by mass spectrometry. The acid sites of nanocomposites MOS/PtO*_x_* was qualitatively characterized by DRIFT spectroscopy of ammonia adsorption. Figure 5 shows DRIFT spectra of materials exposed to NH_3_ (200 ppm) at room temperature. The positive peak evolved at 3100–3200 cm^−1^ due to stretching N−H vibrations, at 1260 cm^−1^ and 1610 cm^−1^ due to symmetric and asymmetric bending NH_3_ vibrations, respectively, and at 1460 cm^−1^ due to symmetric bending NH_4_^+^ vibrations of adsorbed ammonia [31]. The negative band above 3600 cm^−1^ in the stretching O–H vibration range could be due to the depletion of OH−groups (Broensted acid sites) as a result of ammonia adsorption. The positions and intensities of peaks of adsorbed NH_3_ and NH_4_^+^ species were close for pristine MOS and nanocomposites MOS/PtO*_x_*.

### 3.3. A Comparison of Sensitivity to Benzene Vapor

Figure 6a shows the dynamic response to increasing concentration of benzene vapor on the example of pristine and PtO*_x_*−modified WO_3_ sensors. The responses of other sensors are in Supplementary data (Appendix A). For all the samples, the resistance of MOS/PtO*_x_*-based sensors was higher by an order of magnitude than the resistance of MOS sensors. It can be explained by electronic donation from *n*-type MOS to the supported catalytic nanoparticles, probably due to higher work function of PtO*_x_*. It agrees with the XPS data for TiO_2_- and WO_3_-based materials. The sensors signals are plotted in relation to target gas concentration in Figure 6b for the WO_3_-based sensors, and the data for the other MOS and MOS/PtO*_x_* sensors are in Supplementary data (Appendix A). The plots of sensor signals vs. benzene concentration were linear in logarithmic axes fitting the relation *S* ~ *C*^α^. The power α varied in the range α = 0.6–0.9 depending on MOS: α = 0.6 for In_2_O_3_ and TiO_2_, α = 0.7 for ZnO, α = 0.8 for SnO_2_, and the higher value α = 0.9 was characteristic of the WO_3_-based sensors. No effect of the additive PtO*_x_* on the value of the power α was observed. Temperature plots of the sensors signals to 2 ppm of benzene are compared in Figure 6c. The sensitivity increased with temperature, and the highest sensitivity was observed at 200–220 °C which is the maximum limit of operation temperature for PtO*_x_*−modified sensors. MOS/PtO*_x_* sensors demonstrated higher sensitivity to benzene, in comparison to pristine MOS. In Table 3, the sensing performance in the detection of benzene is compared with data in the literature. The presently investigated materials WO_3_/PtO*_x_* and SnO_2_/PtO*_x_* demonstrated a promising sensing behavior: the relatively high sensor signals of S ≈ 5.7–7.8 were registered at a low benzene concentration of 2 ppm. An advantage of the present WO_3_/PtO*_x_* and SnO_2_/PtO*_x_* sensors is the low operation temperature (200–220 °C), in comparison with the higher operation temperatures above 300 °C that have often been applied for metal–oxide-based benzene sensors.

For the sake of comparison, the sensor signals were normalized per the materials’ BET area and the effective sensor signal was determined according to Equation (2):*S*_eff_ = *S*/S_BET_ × 50 m^2^/g (2)

A conditional unit of 50 m^2^/g was used for normalization as an average specific surface area of the investigated samples. Figure 7 shows the highest effective sensor signals, *S*_eff_, to 2 ppm benzene registered at 200–200 °C for different sensing materials in relation to metal–oxygen bond energy of the oxides, which was used as a descriptive parameter. For pristine MOS, the sensitivity increased with *E*_M-O_ in metal oxides from ZnO to SnO_2_ and was almost the same for SnO_2_, TiO_2_, and WO_3_. Another trend was found in the sensitivity of MOS/PtO*_x_* in relation to *E*_M-O_ of the supporting metal oxide. The sensitivity of PtO*_x_*−modified samples steadily increased with metal-oxygen bond energy in the entire range of investigated MOS, i.e., from ZnO- to WO_3_-based nanocomposites.

To highlight the difference in benzene-sensing routes by pristine and PtO*_x_*−modified MOS, the sensitivity was plotted against active sites concentrations (Figure 8). The underlying hypothesis is that sensor signal is directly proportional to the concentration of active sites responsible for target gas conversion. It follows from mass action law applied to a generalized reaction in a stationary state Equation (3):G + <AS>^m−^ → P + m *e*^−^,
*S* = (R_a_ − R_g_)/R_g_ = (σ_g_ − σ_a_)/σ_a_ ~ Δ*n* ~ *C*^α^∙*N*_AS_^β^,(3)
where G is a target gas; <AS> is an active site of target gas conversion with a localized negative charge; P is the product of target gas conversion; m *e*^−^ is a number of released electrons; *N*_AS_ is concentration of active sites; σ_g_ and σ_a_ are electric conductance in presence of target gas and in air, respectively; Δ*n* is change of free electrons concentration as a result of target gas conversion; *C* is target gas concentration; and α and β are rate orders by target gas and active site, respectively. Through compilation of the data of effective sensor signals (Figure 7) and active sites concentrations at the surface of different samples (Figure 4), two relations of sensitivity vs. active sites concentrations were distinguished. The one shown in Figure 8a is the dependence of sensitivity of pristine MOS on the summary concentrations of oxidative sites (*N*_H2,TPR_ measured by TPR) and acid sites (*N*_acid sites_ estimated by TPD) at the surface of MOS. The sum of concentrations of both types of active sites in the *x*−axes was chosen to satisfy the initial assumption that sensitivity is proportional to the number of acid sites Equation (3). Figure 8b shows the plot of MOS/PtO*_x_* sensitivity vs. acid sites concentration at the material surfaces. In this case, when the concentration of oxidative sites at the surface of MOS/PtO*_x_* was summed with that of acid sites, no correlation with the sensitivity was found.

### 3.4. DRIFT Study of Material Interaction with Benzene Vapor

DRIFT spectra were registered during the materials exposure to benzene vapor at room temperature to characterize the adsorption and at raised temperature (220 °C) to investigate the redox conversion of target molecules. The pattern of adsorbed C_6_H_6_ onto the surface of MOS was observed by FTIR spectroscopy of metal oxides impregnated with liquid benzene and dried at room temperature (Figure 9a). It demonstrates the peaks of aromatic ring vibrations at 1480 cm^−1^, stretching C=C vibrations at 1580–1540 cm^−1^, stretching C-H vibrations at 3040–3100 cm^−1^, and overtones at 1800 cm^−1^ and 1960 cm^−1^. DRIFT spectra of pristine MOS and PtO*_x_*−modified MOS exposed to benzene vapor at room temperature are shown in Figure 9b. The peak of aromatic ring vibrations at 1480 cm^−1^ and the band of stretching C=C vibrations at 1580–1540 cm^−1^ in adsorbed C_6_H_6_ were observed on the DRIFT spectra of TiO_2_− and WO_3_−based materials. The more intense peak of polar C=O group stretching vibrations evolved at 1670 cm^−1^ on the spectrum of pristine WO_3_ exposed to benzene (Figure 9b). This peak, in combination with the stretching C=C vibrational band at 1580–1540 cm^−1^, is typical of *o*-benzoquinone [32], and its appearance indicates a partial oxidation of adsorbed benzene at the surface of tungsten oxide. Figure 9c shows DRIFT spectra of nanocomposites MOS/PtO*_x_* exposed to benzene at room temperature. The evolved peaks were similar to those on the spectra of pristine MOS (Figure 9b). Additionally, on the spectrum of WO_3_/PtO*_x_*, two minor positive peaks appeared at 2100 cm^−1^ and 1860 cm^−1^; this can be due to trace amounts of adsorbed CO resultant from a catalytic oxidation of benzene, as discussed below. The minor negative IR absorption bands at 1540–1560 cm^−1^ and 1260–1300 cm^−1^ on the spectra of ZnO− and In_2_O_3_−based samples can be attributed to bidentate carbonate species [33] which exist at the surface of basic oxides due to CO_2_ adsorption and can be partially removed in the process of interaction with the target gas molecules.

DRIFT study of benzene conversion on pristine MOS at temperature 220 °C showed the prominent spectral on the spectrum of WO_3_ (Figure 10a). The peaks at 1670 cm^−1^ and 1560 cm^−1^ evolved, suggesting benzoquinone formation from the oxidation of benzene. Simultaneously, the W–OH band at 1410 cm^−1^ and the O–H stretching vibrational band at 3200–3600 cm^−1^ decreased. Thus, oxidation of benzene at the surface of WO_3_ involved reactive surface oxygen species and OH-groups Equation (4):C_6_H_6(g)_ + 3/n O_n_^m-^_(surf)_ → C_6_H_4_O_2(ads)_ + H_2_O_(g)_ + 3m/n *e*^−^(4)

DRIFT spectra of PtO*_x_*−functionalized MOS exposed to benzene at 220 °C displayed the peak of in-ring C=C vibrations (1540 cm^−1^), like in adsorbed benzene or benzoquinone; the peak of C=O stretching vibrations (1670 cm^−1^) characteristic of benzoquinone was weak (Figure 10b). The main distinctive feature on the spectra of PtO*_x_*−functionalized MOS was the evolution of C≡O vibrational peak at 2100–2140 cm^−1^, like in adsorbed CO [33]. In comparison with the benzoquinone-like C=O peak at 1670 cm^−1^, the peak of Pt-bound CO (2140 cm^−1^) was more intense (Figure 10b). The minor peak which evolved at 1860 cm^−1^ could be attributed to Pt−bound CO in a bridging conformation, and the stronger peak at 2140 cm^−1^ is typical of terminal Pt–CO groups [33]. Thus, in the presence of a PtO*_x_* catalyst, the oxidation of adsorbed benzene results in the formation of CO according to reaction 5:C_6_H_6(g)_ + 9/n O_n_^m-^_(surf)_ → 6 CO_(ads)_ + 3 H_2_O_(g)_ + 9m/n *e*^−^(5)

The evolution of Pt–CO species was observed by DRIFT spectroscopy for nanocomposites based on acidic MOS: SnO_2_, TiO_2_, and WO_3_ (Figure 10b). The intensity of carbonyl peaks increased in the same order, corresponding to the increment of metal–oxygen bond energy and surface acidity of MOS.

## 4. Discussion

This work is a comparative and systematic study of active sites and sensitivity to benzene of *n*−type wide bandgap MOS that are often used as VOC sensors: ZnO, In_2_O_3_, SnO_2_, TiO_2_, and WO_3_ [1,3,4,5,6]. These oxides differ widely in the fundamental parameters such as cationic charge/radius ratio, metal–oxygen bond energy, and formation heat (Table 1). It is of interest to uniformly compare surface reactivity and sensing behavior to benzene of pristine MOS and of the oxides modified by catalytic PtO*_x_* nanoparticles. In order to obtain the comparable microstructures of sensing materials, the samples of nanocrystalline MOS were obtained under similar synthetic conditions by the aqueous chemical deposition route followed by thermal treatment of deposited metal hydroxides. Although the materials had similar morphologies of irregularly shaped agglomerated nanoparticles (Figure 1), the microstructural parameters of the obtained MOS differed in definite ranges: crystallite size 5–50 nm and BET surface area 5–100 m^2^/g (Table 2). This results from distinct energetics and kinetics of crystallization of metal oxides and complicates the comparison of sensing behavior because the sensitivity depends on the microstructure of a sensing material [34,35]. In addition, the elevated annealing temperature of titania (700 °C), relative to that of other MOS (300 °C), was used to lower the intrinsically high electric resistance of TiO_2_-based sensors which could not have been measured otherwise.

PtO*_x_* nanoparticles with the size 5–10 nm (Figure 1e,f) were immobilized onto the surface of MOS via the impregnation technique. The absence of diffraction peaks from PtO*_x_* phases in XRD (Appendix A) and ED (insets in Figure 1a–d) patterns could be due to the low percentage of the additive and/or its poor crystallinity. By XPS, it was shown that Pt^2+^ was the predominant oxidation state of the catalytic additive. The oxidized state of platinum could be due to adsorbed oxygen on surface platinum atoms, provided that the small PtO*_x_* nanoparticles have high surface-to-volume ratio. A minor fraction of reduced Pt^0^ species was found in the nanocomposites TiO_2_/PtO*_x_* and WO_3_/PtO*_x_*. The presence of reduced platinum atoms in these nanocomposites may be attributed to the electronic interaction of PtO*_x_* nanoparticles with the supporting oxides, as follows from the shifts of XPS peaks of Ti^4+^, W^6+^, and O^2−^ relative to pristine MOS (Figure 2a,b). That the electronic donation was observed only from TiO_2_ and WO_3_ can be explained by the higher concentration of oxygen vacancies in these oxides, compared with ZnO, In_2_O_3_, and SnO_2_. The single changed spin centers V_O_^∙^ were determined by EPR in similar MOS samples in our previous works, as summarized in the recent review [36]. The higher oxygen deficiency of tungsten oxide was also evidenced by the occurrence of W^5+^ state in WO_3_ (Figure 2a), and the concentration of W^5+^ was estimated to be 5 at.% of the total W content in WO_3_/PtO*_x_*, i.e., twice as low as in pristine WO_3_. Titania is also prone to oxygen deficiency, as was shown by the occurrence of Ti^3+^ state in EPR spectra of nanocrystalline TiO_2_ [37]. There is a series of stable oxygen-deficient homologs Ti_n_O_2n−1_ (n = 3–10, 20) in which Ti^3+^ and Ti^4+^ states coexist [38]. Numerous WO_3-δ_ (δ = 10^−2^–10^−1^) phases, including Magneli series, are known in the W–O system between stoichiometric WO_2_ and WO_3_ [39]. It demonstrates the stronger tendency of WO_3_ and TiO_2_ to be oxygen-deficient, in comparison with ZnO, In_2_O_3_, and SnO_2_ which have only limited oxygen deficiency (of the order δ ≈ 10^−6^–10^−3^ in MO*_x_*_−δ_) depending on temperature [40,41,42].

Two types of active sites at the materials surfaces were analyzed: oxidative sites and acid sites. Oxidative sites include all surface species that can possess an oxidative reactivity, e.g., lattice and chemisorbed oxygen species and/or hydroxyl groups. To evaluate the oxidative sites, the materials were characterized by temperature-programmed reduction with hydrogen (TPR). The oxidative surface sites can participate in the sensing process of reducing gases such as benzene because the gas-solid interaction proceeds at the sensor surface. The estimated concentrations of oxidative sites (Figure 4a) were close at the surfaces of SnO_2_ and TiO_2_ and exceeded those at the surfaces of ZnO, In_2_O_3_, and WO_3_. This is in agreement with our previous results [24] and can be explained by a proper balance between oxygen chemisorption and oxygen vacancy formation favored by the intermediate metal–oxygen bond energies in SnO_2_ and TiO_2_ [23,36]. The concentrations of oxidative sites on these samples are comparable to the monolayer of surface atoms of approximately 10^19^ m^−2^/6.02∙10^23^ mole^−1^ ≈ 1.6∙10^−5^ mole/m^2^. Lower surface reducibility was observed for the oxides with relatively lower *E*_M-O_ (ZnO, In_2_O_3_) and higher *E*_M-O_ (WO_3_), likely because of imbalance in the interplay between O_2_ chemisorption and desorption of lattice oxygen. The estimated concentrations of oxidative sites were higher at the surfaces of PtO*_x_*−modified oxides, as compared with pristine MOS, but the general trend in relation to metal–oxygen bond energy was the same (Figure 4a). This may be due to the catalytic (spillover) effect of PtO*_x_* nanoparticles, which facilitates oxygen exchange between gas phase and supporting oxide surface [43].

Acid sites of the Broensted-type (acidic OH-groups) and Lewis-type (coordinately unsaturated surface cations) were determined by TPD of ammonia. Lewis acid sites play a crucial role in the adsorption of gas molecules having lone-electron-pair donor atoms which contain, for example, nitrogen or oxygen. Therefore, quantification of acid sites is helpful to understanding the sensing behavior to target gases with Lewis base properties. As follows from Figure 4b, surface acidity of MOS increased in the order ZnO ≈ In_2_O_3_ < SnO_2_ < TiO_2_ < WO_3_, which agrees with the increment of metal–oxygen bond energy. The origin of this correlation should be the increase of cationic charge/radius ratio (Table 1): the larger the positive charge density at the cation, the stronger its lone-pair acceptor behavior (Lewis acidity) and the higher the ionic bond energy with oxide anions. Because in TPD method the use of elevated temperature is inherent, the surface acidity of catalytically PtO*_x_*−modified materials could not be measured by this method due to catalytic oxidation of probe molecules prior to desorption. To overcome this difficulty, we used DRIFT spectroscopy to estimate the surface acidity of nanocomposites MOS/PtO*_x_*. The DRIFT spectra (Figure 5) were registered under the conditions of NH_3_ adsorption at room temperature so that no catalytic oxidation was interfering. The observed bending vibrational peaks of NH_4_^+^ and NH_3_ surface species can be used as characteristics of ammonia adsorption on Broensted (OH-groups) and Lewis acid sites (surface cations), respectively. Although the quantitative information could not be obtained from DRIFT spectra, the peak intensities of adsorbed NH_4_^+^ and NH_3_ species were close for pristine MOS and nanocomposites MOS/PtO*_x_*. This provides evidence that the relative concentrations of acid sites at the oxides surfaces were unaffected by the supported PtO*_x_* nanoparticles, likely due to low percentage (1 wt.%) of the additive.

The materials were sensitive to 0.5–5 ppm of benzene vapor in air. In comparison with pristine MOS, the PtO*_x_*−modified sensors had an improved sensitivity to benzene in the entire tested ranges of target gas concentration (Figure 6b) and operation temperature (Figure 6c). This is explained by the catalytic effect of metal–oxide-supported platinum in the oxidation of organic molecules [18,44]. In order to reveal the effect of materials composition and active sites concentrations on the sensitivity, we introduced the effective sensor signal defined by Equation (2). This is an attempt to uniformly compare the sensitivity of materials with different chemical compositions without regard to the influence of distinct microstructure parameters (Table 2). It was commonly noticed that sensor signals to reducing gases were linear if plotted against the surface-to-volume ratio of a sensing material, i.e., inversely proportional to an average particle size or directly proportional to specific surface area [34,35]. This dependency was theoretically substantiated by that which was reported by Rothschild and Komem, (2004) [45]. Comparing the trends in active site concentrations (Figure 4) and in the sensitivity to benzene (Figure 7) in relation to metal–oxygen bond energy, one can suggest distinct impacts from the active sites to the sensing process in the absence and in the presence of PtO*_x_* catalyst. The trend in effective sensor signals to benzene of pristine MOS in relation to *E*_M-O_ is similar to that previously observed for the sensitivity of *n*−type MOS to methanol and acetone [24]. This assumes that the sensitivity to these different classes of organic molecules is controlled by similar factors.

Different correlations between the sensitivity to benzene and the concentrations of active sites were deduced for pristine MOS and nanocomposites MOS/PtO*_x_*. For pristine MOS, comparison of the plots of sensitivity (Figure 7) and active sites concentrations in relation to *E*_M-O_ (Figure 4) led to an intuitive conclusion that oxidative sites and acid sites were equivalently responsible for the conversion of benzene. Plotted against the total concentration of oxidative and acid sites, the sensitivity to C_6_H_6_ continuously increased in the entire range (Figure 8a). A similar result was previously obtained for the sensitivity of *n*-type MOS to acetone and methanol [24,36]. The role of oxidative sites is evidently the oxidation of VOC target molecules, and acid sites can influence the adsorption of Lewis base molecules possessing donor oxygen atoms (e.g., alcohols or ketones). For the conversion of benzene, the role of acid sites is likely binding the conversion product (benzoquinone) rather than the target molecules (DRIFT results in Section 3.4). For PtO*_x_*−modified sensors, the graph of sensitivity to benzene vs. *E*_M-O_ (Figure 7) resembles that of acid site concentration at the surface of MOS (Figure 4b). Supported by the DRIFT study of ammonia adsorption at room temperature (Figure 5), it was assumed that the relative acidity of modified MOS was unaffected in the presence of a low percentage (1 wt.%) of the additive PtO*_x_*. The steady growth of MOS/PtO*_x_* sensitivity with the concentration of acid sites suggests that surface acidity is of primary importance for the efficient conversion of benzene in the presence of PtO*_x_* catalyst.

Using DRIFT spectroscopy, the distinction in sensing mechanisms of pristine and PtO*_x_*−modified MOS was revealed. DRIFT spectra demonstrated minor traces of adsorbed benzene. This could be due to low concentration of target gas (10 ppm), as well as low activity of infrared vibrations of non-polar C–C and weakly polar C–H bonds in C_6_H_6_ molecules. The adsorption of C_6_H_6_ could be observed on the acidic oxides TiO_2_ and WO_3_ (Figure 9b) along with the product of partial benzene oxidation: *o*-benzoquinone C_6_H_4_O_2_. The molecules of benzoquinone possess polar C=O groups which have a highly active infrared vibrational mode at 1670 cm^−1^. The modification of MOS by catalytic PtO*_x_* nanoparticles did not affect the adsorption of benzene, as follows from the evolution of the same peaks on the room temperature DRIFT spectra of MOS (Figure 9b) and MOS/PtO*_x_* (Figure 9c). At raised temperature (220 °C), simulating the conditions of sensors operation, benzene molecules were partially oxidized, yielding *o*-benzoquinone Equation (4). The stronger DRIFT signals observed for WO_3_-based samples correlate with the higher sensitivity to benzene. The high sensitivity and surface reactivity to benzene can be rationalized by enhanced surface acidity of tungsten oxide. Lewis acid sites, i.e., coordinately unsaturated cations, act as adsorption sites for benzoquinone molecules via the lone-pair donor O-atoms. This favors the oxidation reaction Equation (4) through binding the product of benzene oxidation. At the surface of PtO*_x_*−modified MOS at raised temperature (220 °C), oxidation of benzene results in the formation of CO (Equation (5)), i.e., proceeds deeper than on pristine MOS. It can be the reason for higher sensitivity of nanocomposites MOS/PtO*_x_* to benzene. However, this route of PtO*_x_*−catalyzed benzene oxidation was strongly dependent on the surface acidity of supporting MOS. The intensity of the evolved peaks of Pt-bound CO on the DRIFT spectra of MOS/PtO*_x_* (Figure 10b) steadily increased with the increment of oxides surface acidity in the order ZnO ≈ In_2_O_3_ < SnO_2_ < TiO_2_ < WO_3_. This agrees with the increasing sensitivity of MOS/PtO*_x_* sensors to benzene (Figure 8b).

A similar effect of the acidity of supporting oxides was observed for catalytic activity of supported platinum in the combustion of hydrocarbons [18,46]. It was argued that zero-valent Pt^0^ atoms were the key catalytically active sites. The appearance of reduced Pt^0^ species in the catalytic cycle was favored on acidic supports because basic oxides stabilized oxidized platinum in the form of platinates [46]. In the present work, the occurrence of reduced Pt^0^ species in PtO*_x_*−modified acidic oxides TiO_2_ and WO_3_ was confirmed by XPS (Figure 2c). Thus, surface acidity of MOS plays a dual role in improving the sensitivity of MOS/PtO*_x_* nanocomposites to benzene. On the one hand, it facilitates the occurrence of Pt^0^ sites which catalyze deep benzene oxidation to CO. On the other hand, acid sites favor the adsorption of benzoquinone which is a by-product or an intermediate in the catalytic benzene oxidation. The catalytic impact of PtO*_x_* nanoparticles facilitated by surface acidity of supporting MOS is likely more important in determining the sensitivity of MOS/PtO*_x_*, respective of the role of native oxidative sites (surface oxygen species, and hydroxyls) in the oxidation of benzene at the MOS surfaces. This may be the reason for the independence of nanocomposite sensitivity on the concentration of oxidative sites (Figure 8b).

## 5. Conclusions

Active surface site concentrations and sensitivity to benzene were compared for pristine and PtO*_x_*-modified *n*-type metal oxide semiconductors. MOS and MOS/PtO*_x_* materials showed different trends in the sensitivity to benzene in relation to metal–oxygen bond energy and active site concentrations, related to distinct sensing mechanisms. The higher sensitivity of pristine SnO_2_, TiO_2_, and WO_3_ was enabled by equivalently important roles of acid sites and oxidation sites in the processes of benzene conversion and adsorption of the resultant benzoquinone. Modification by PtO*_x_* improved the sensitivity to benzene due to catalytic deep oxidation of target molecules to carbonyl species. The sensitivity of PtO*_x_*−modified MOS increased with the surface acidity of supporting oxide. The synergistic effect of surface acidity and PtO*_x_* catalytic activity on the sensitivity of nanocomposites MOS/PtO*_x_* to benzene was rationalized by the stabilization of zero-valent Pt species, which is critical for the deep oxidation of benzene to CO. These things considered, acid sites favor benzene oxidation through binding benzoquinone which is one of the target gas oxidation products.

## Figures and Tables

**Figure 1 sensors-22-06520-f001:**
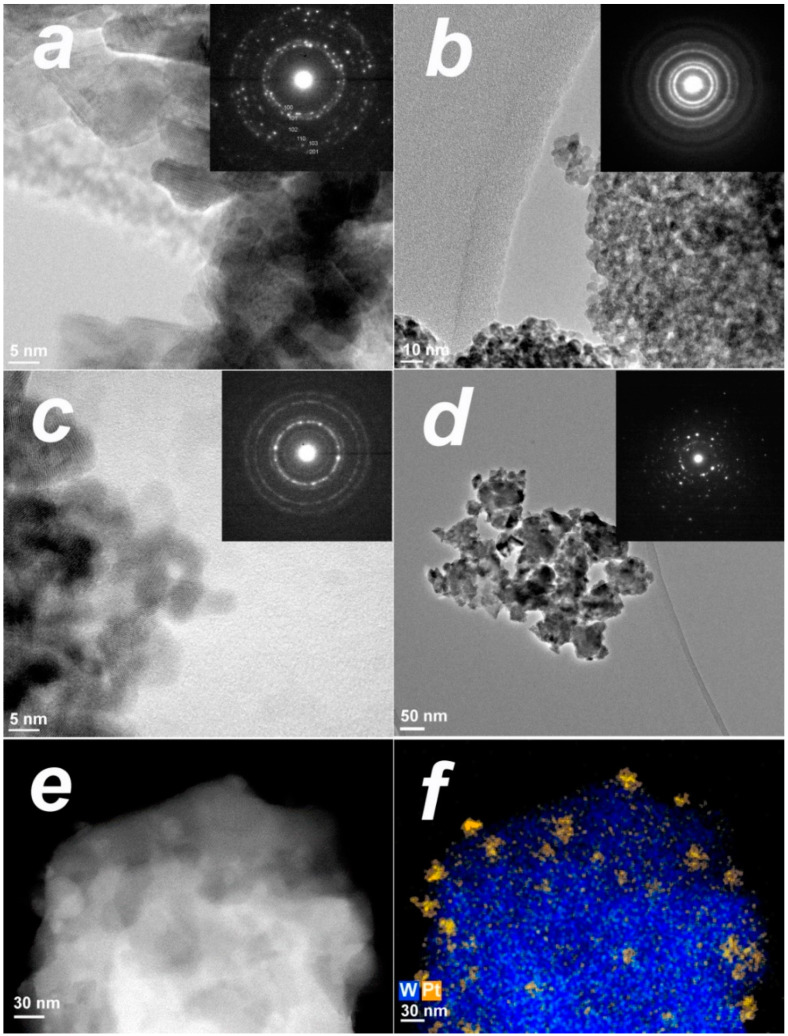
HRTEM images of nanocomposites ZnO/PtO*_x_* (**a**), SnO_2_/PtO*_x_* (**b**), and In_2_O_3_/PtO*_x_* (**c**); TEM image of TiO_2_/PtO*_x_* (**d**); HAADF-STEM micrograph of WO_3_/PtO*_x_* (**e**), and overlaid EDX map of W L-signal (blue) and Pt L-signal (orange) (**f**).

**Figure 2 sensors-22-06520-f002:**
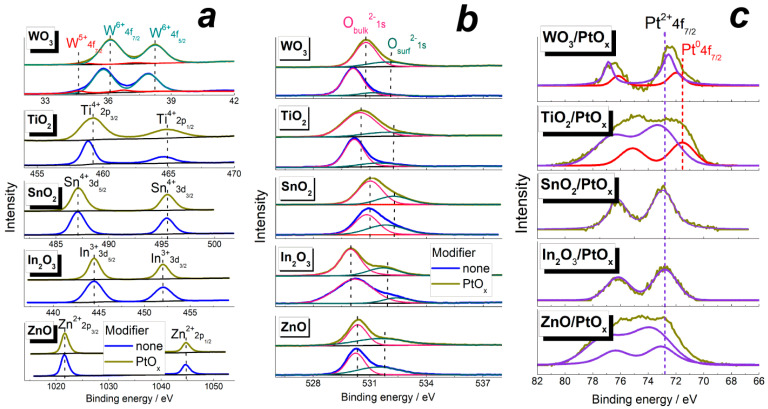
High-resolution XP-spectra of MOS and MOS/PtO*_x_* materials in the characteristic binding energy limits of metals constituting MOS (**a**), O 1s (**b**), and Pt 4f (**c**).

**Figure 3 sensors-22-06520-f003:**
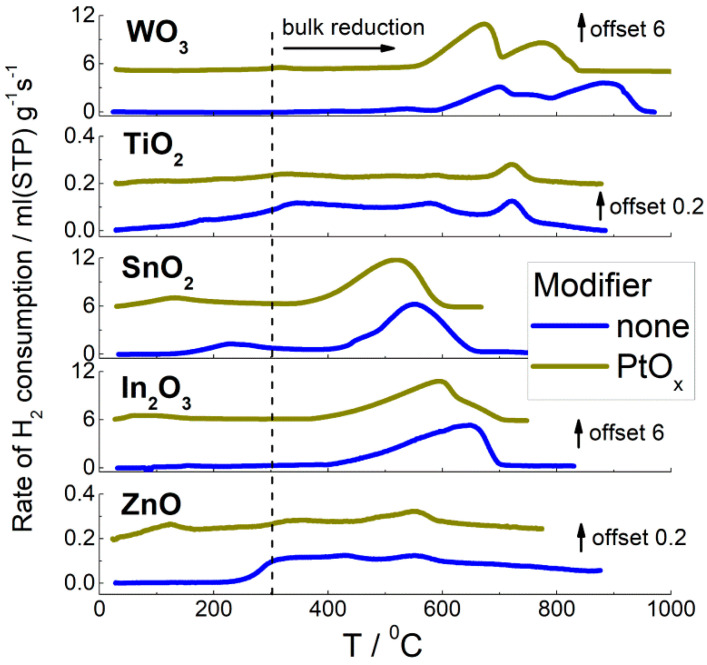
TPR patterns of pristine and PtO*_x_*−modified metal–oxide semiconductors.

**Figure 4 sensors-22-06520-f004:**
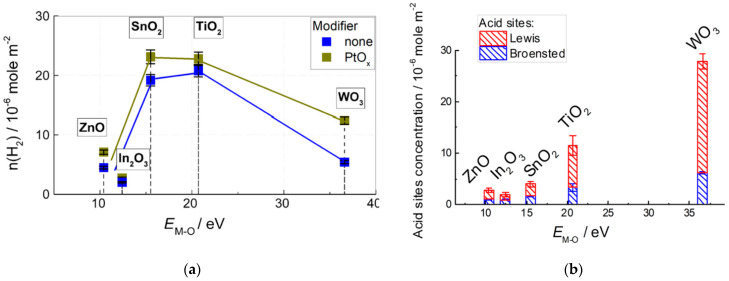
Concentration of oxidative sites at the surface of pristine and PtO*_x_*−modified MOS in equivalent amount of consumed H_2_ for surface species reduction in TPR (**a**) and concentration of Broensted and Lewis acid sites at the surface of MOS determined by TPD of ammonia (**b**) in relation to metal–oxygen bond energy in MOS.

**Figure 5 sensors-22-06520-f005:**
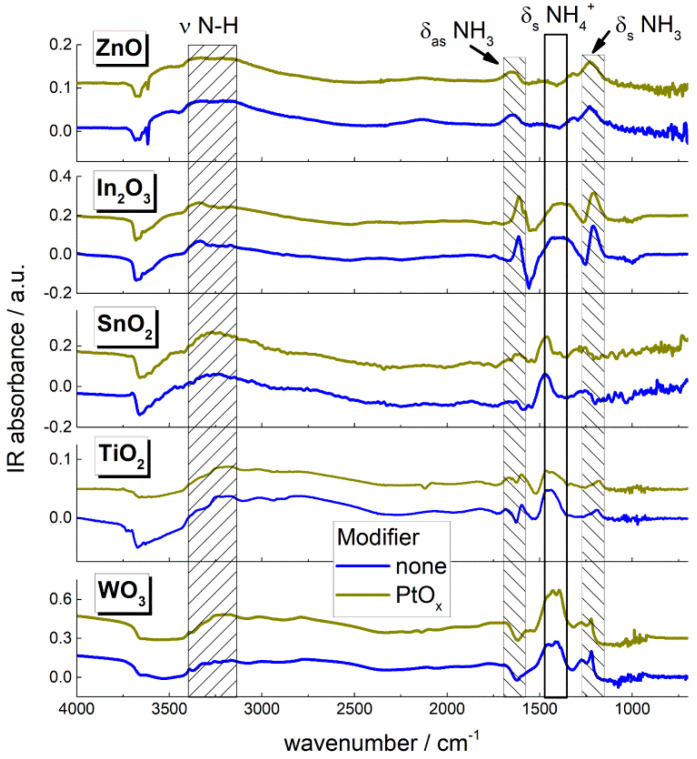
DRIFT spectra of pristine and PtO*_x_*−modified MOS exposed to 200 ppm NH_3_ at room temperature for 1 h. Positions of stretching (ν N-H) and symmetric (δ_s_) and asymmetric (δ_as_) vibrations of adsorbed NH_3_ and NH_4_^+^ species are shown.

**Figure 6 sensors-22-06520-f006:**
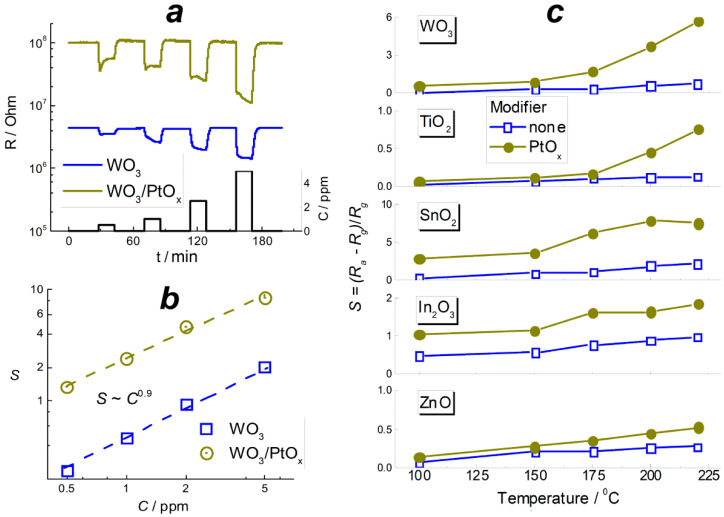
Dynamic response of WO_3_ and WO_3_/PtO*_x_* sensors to increasing concentration of benzene at 220 °C (**a**); dependence of sensor signals of WO_3_ and WO_3_/PtO*_x_* sensors on C_6_H_6_ concentration at 220 °C (**b**); sensor signals of pristine and PtO*_x_*−modified MOS to 2 ppm C_6_H_6_ in relation to temperature (**c**).

**Figure 7 sensors-22-06520-f007:**
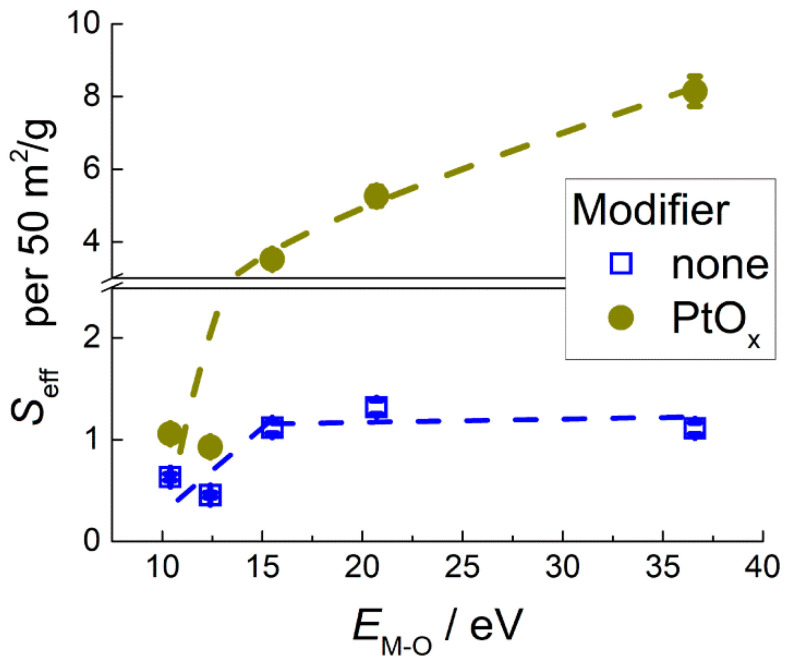
Maximum effective sensor signals of pristine and PtO*_x_*−modified MOS to 2 ppm C_6_H_6_ in relation to metal–oxygen bond energy in MOS.

**Figure 8 sensors-22-06520-f008:**
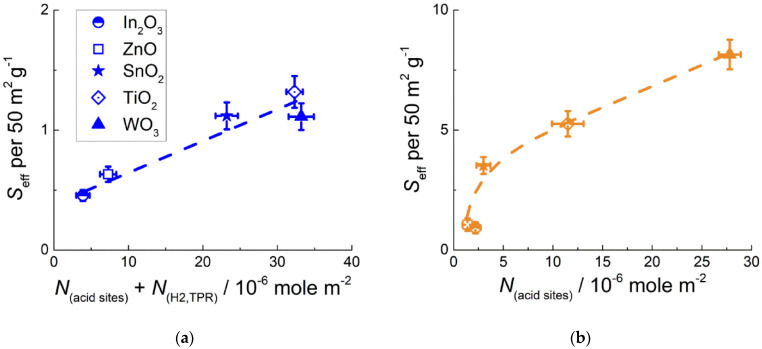
Maximum effective sensor signals to 2 ppm C_6_H_6_ of: (**a**) pristine MOS in relation to summary concentration of acid sites and oxidative sites (in equivalents of H_2_ consumed in TPR) and (**b**) PtO*_x_*−modified MOS in relation to concentration of acid sites at the surface of MOS.

**Figure 9 sensors-22-06520-f009:**
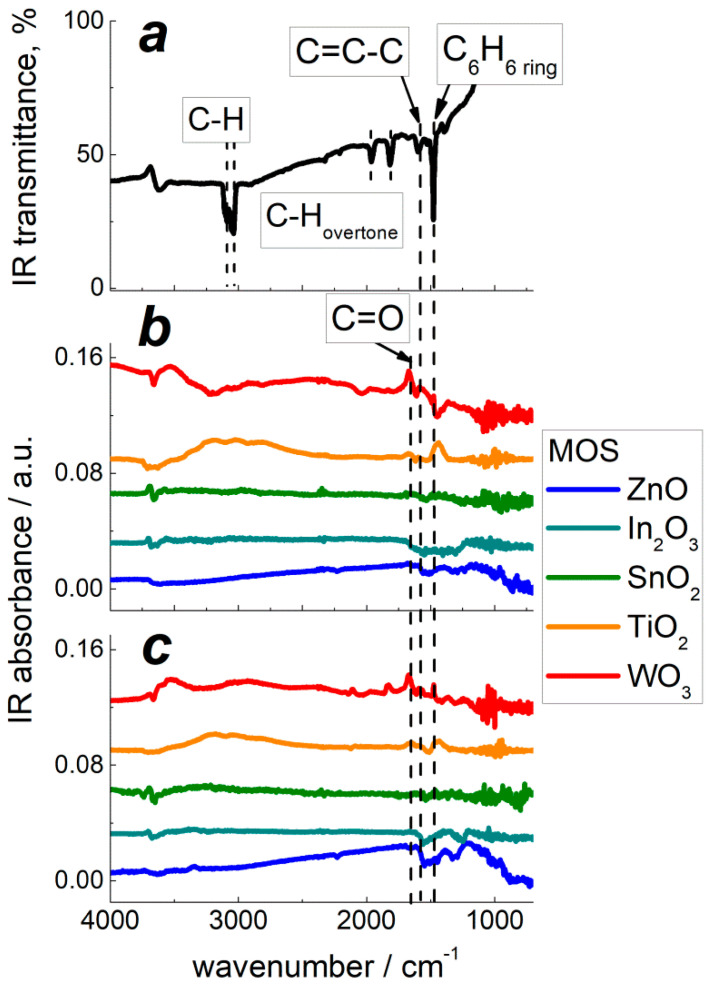
FTIR spectrum of SnO_2_ impregnated with liquid C_6_H_6_ (**a**); DRIFT spectra of pristine MOS (**b**), and PtO*_x_*−modified MOS (**c**) exposed to 10 ppm C_6_H_6_ for 1 h at room temperature.

**Figure 10 sensors-22-06520-f010:**
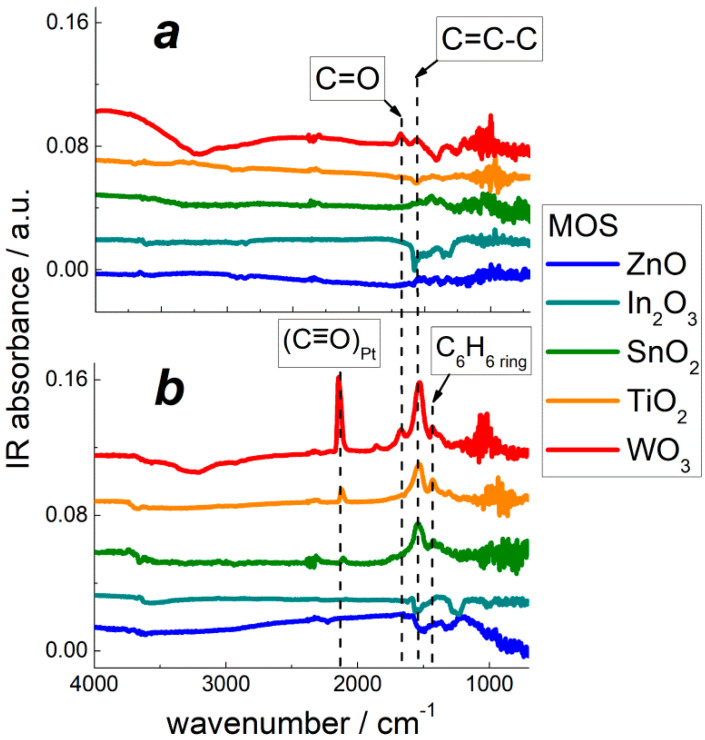
DRIFT spectra of pristine MOS (**a**) and PtO*_x_*−modified MOS (**b**) exposed to 10 ppm C_6_H_6_ for 1 h at 220 °C.

**Table 1 sensors-22-06520-t001:** Parameters of metal–oxide semiconductors used in the work: r(M*^n^*^+^)—effective cationic radii coordinated by oxide anions [25], metal–oxygen bond energy *E*_M-O_, formation enthalpy Δ_f_H°(M*_x_*O*_y_*) per metal atom, and band gap energy (*E*_g_) [26,27,28].

MOS	r(M*^n^*^+^), Å	*E*_M-O_^a^, eV	−1/*x*·Δ_f_H° (M*_x_*O*_y_*), eV	*E*_g_, eV
ZnO	0.60	10.4	3.6	3.4
In_2_O_3_	0.79	12.4	4.8	2.8
SnO_2_	0.69	15.5	6.0	3.6
TiO_2_	0.61	20.7	9.8	3.0–3.2
WO_3_	0.58	36.6	8.7	2.5–2.8

^a^ Bond energy in metal oxides calculated from thermodynamic parameters [26]: *E*_MO_ = {−Δ_f_H°(M_x_O_y_) + x·Δ_sub_H°(M) + y/2·Δ_dis_H°(O_2_) + y·Δ_el.af._H°(O) + ΣΔ_ion._H°(M)}/(y·CN_O_), where Δ_sub_H°(M)—sublimation enthalpy of metal, Δ_dis_H°(O_2_)—dissociation enthalpy of oxygen molecule, Δ_el.af._H°(O)—electron affinity of oxygen atom, ΣΔ_ion._H°(M)—sum of ionization potentials of metal cation, CN_O_—coordination number of oxygen in the oxide.

**Table 2 sensors-22-06520-t002:** Composition of samples, annealing temperature of MOS (T_anneal_), crystallite size (*d*_XRD_), and BET area (*S*_BET_).

MOS	T_anneal_ (°C)	Additive	Phase Composition	*d*_XRD_ (nm)	*S*_BET_ (m^2^/g)
ZnO	300	none	ZnO wurtzite	18–20	18–21
		PtO*_x_*			
In_2_O_3_		none	In_2_O_3_ cubic	8–10	103–110
		PtO*_x_*			
SnO_2_		none	SnO_2_ tetragonal	4–6	95–100
		PtO*_x_*			
WO_3_		none	γ−WO_3_ monoclinic	9–12	32–35
		PtO*_x_*			
TiO_2_	700	none	TiO_2_ anatase (33 mol.%), TiO_2_ rutile (67 mol.%)	27–30 (anatase) 38–46 (rutile)	7–8
		PtO*_x_*			

**Table 3 sensors-22-06520-t003:** Composition of sensing behavior to benzene of the presently investigated resistive sensors and literature data.

Material	Morphology	Benzene Concentration, ppm	Sensor Signal, *S* = (*R_a_* − *R_g_*)/*R_g_*	Operation Temperature, °C	Ref.
SnO_2_/Cu_2_O	nanowires	10	11.5	300	[11]
ZnO/Pt	nanowires	10	0.05	100	[12]
WO_3_/Al_2_O_3_, Pt	nanoparticles	1	0.97	250	[13]
SnO_2_/TiO_2_, Rh	hollow spheres	5	80	325	[14]
SnO_2_/Pd	nanowires	1	24.5	300	[15]
SnO_2_/Pt			7.3	300	
WO_3_	nanoneedles	1	1	200	[16]
SnO_2_/PtO*_x_*	nanoparticles	2	7.8	200	this work
WO_3_/PtO*_x_*			5.7	220	

## Data Availability

Not applicable.

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
