# Peer review of "Synergistic Effect of Surface Acidity and PtO*_x_* Catalyst on the Sensitivity of Nanosized Metal–Oxide Semiconductors to Benzene"

_sensors, 2022, doi:10.3390/s22176520_

Round 1
Reviewer 1 Report
1. How is the informaiton obtained from TPR related to sensing activity?
In 3.4 DRIFT Study,
2. There are contradictions in the context and Fig.7 caption. It caused the confusion in the interpretation of the data.
Line 350: The more intense peak of polar C=O group stretching vibrations evolved at 1670 cm-1 on the spectrum of WO3 exposed to benzene (Figure 7b).
Line 354: The presence of catalytic PtOx nanoparticles in the nanocomposites did not affect the adsorption of benzene, as was deduced from the evolution of the same peaks as on the spectra of pristine MOS (Figure 7c).
3. How does the author know tungsten oxide exhibit enhanced surface acidity, i.e., lewis acid sites ?
4. If deep oxidation of benzene occurs in PtOx modified MOS, did the author observe more water singal in IR spectra?
In Conclusion
5. How does the author determine the sensitivity of materials to 0.5–5 ppm of benzene vapor in air correlating with the concentrations of active sites?
Author Response
We thank the Reviewer for consideration of the article and valuable comments and questions. Below are the responses to the questions. We made some corrections in the revised manuscriptaccording to Reviewer comments, the changes are marked by grey marker.
1) Temperature programmed reduction (TPR) is a tool to characterize the oxidative activity of the oxides surfaces, since hydrogen is consumed to some extent at temperature lower than the temperature of complete reduction of oxides to metals. The amount of hydrogen consumed during the reduction of oxidative surface species (or, oxidative sites) is a measure of their concentration at the material surface. These oxidative sites (surface oxygen anions, chemisorbed species of oxygen or hydroxyl groups) can play a key role in the sensing process of reducing gases, since the sensor response is a result of oxidation of the analyzed gas molecules at the material surface. In this case, there should be a direct correlation between the sensitivity to a reducing gas and concetration of oxidative sites. Our goal was to establish if the sensitivity to benzene correlates with the concentrations of active sites at the surfaces of different sensing materials. It turned out that the consideration of the oxidative sites alone (measured by TPR) is not enough to explain the difference in materials sensitivity, and also acid sites (probed by TPD) should be taken into account. More detailed, oxidative sites and acid sites are both important for the sensitivity of pristine MOS. While for PtOx-modified oxides the surface acidity is a more important factor because of the catalyst-support interplay. This was discussed throughout the main text in the article. We added a couple of sentences explaining the motivation to use TPR at the beginning of Discussion section in the revised manuscript.
2) We reformulated the sentences so as to clarify the confusion in the Figure describing DRIFT study of benzene adsorption. The changes are in the revised text.
3) The probe of surface acidty used in the study was temeprature programmed desorption of ammonia. In the revised text, we added the decription of the method in the Materials and Methods section. Briefly, ammonia gas is adsorbed at the pretreated surfaces of oxides, and further it is desorbed to a flow of helium in increasing temperature environment. Adsorbed molecules of ammonia are held weaker on Broensted acid sites and stronger - on Lewis acid sites. So that, measuring the amount of ammonia desorbed at different temperature intervals allows to compare surface acidity of Broented and Lewis types for different samples.
4) In the DRIFT spectra of samples exposed to benzene vapor at temperature 220 C, we did not observe the peaks of water evolved from the oxidation reaction. We suppose that water was eliminated as a vapor in the gas ambient, while the method of diffuse reflectance infrared spectroscopy is focused at the sample surface. So, the adsorbed species at the solid surface are readily detected by DRIFT spectroscopy, while gas phase molecules cannot be detected. The reason for the desorption of water vapor from the samples surfaces should be the elevated temperature in the experiment (220 C).
5) We rephraised this sentence to make it more clear. Herein, we meant two points. 1) The materials showed sensitivity to the indicated range of benzene concentrations in air. 2) The sensitivity (for certainty compared to1 ppm of benzene) was deduced to correlate with the active sites concentrations at the materials surfaces. Sensitivity was determined from resistance measurements and normalized by the BET surface area. Active sites concetrations were determined by TPR (oxidative sites) and TPD (acid sites). These parameters were point-by-point merged and compared for MOS and MOS/PtOx samples in Figure 6 of the original manuscript (Figure 8 in the revised manuscript). The observed increment in sensitivity with the concentrations of oxidative+acid sites (for MOS) or only acid sites (for MOS/PtOx) lead us to the conclusion that the discussed correlations take place.
Reviewer 2 Report
The authors present the study of surface reactivity and sensitivities to benzene of metal oxide semiconductors modified by Pt. The topic is of relevant scientific interest, however, making a very general review of the manuscript, aspects can be observed, which it is suggested to be corrected.
Introduction
Line 57 to 72. In this section, the objective(s) of the work are usually stated based on previously presented information so that the scientific contribution of the work is highlighted. It is necessary to improve and to reduce, because is widely described and it is not necessary present results.
Line 73 to 80. The information in the table should serve to delve deeper into the justification and relevance of the work, because it is only presented and mentioned that the MOS differ in fundamental parameters. Or, it is suggested to switch to section 2. Materials and Methods, because the properties of the semiconductors to be prepared are detailed in the table.
Materials and Methods
Between Line 81 and line 82. It is suggested to put subtitle 2.1. Sensor Preparation
Between Line 98 and line 99. It is suggested to put subtitle 2.2. Sensor Characterization
Between Line 125 and line 126. It is suggested to put subtitle 2.3. Evaluation of Sensitivity of Sensor
Line 107-109. It is suggested to change micrographs for microscopy, because the technique is the microscopy and not micrographs
Line 133. It is necessary to change the “2,0-3,5” by “2.0-3.5”
Line 140. It is suggested to complement ….according to equation 1
Line 141. It is suggested to change … Rg – resistance in test gas, by …where Rg is resistance in test gas.
In this section is necessary to describe the characterization for TPD because because the results are being presented.
Results and Discussion
Line 148-149. not exist this document with Supplementary data.
Line 166. It is necessary to change the (Fig. 1e,f) by (Figure 1e,f)
Line 176 and 208. It is necessary to change the (Fig. 2a) by (Figure 2a)
Line 182. It is necessary to change the (Fig. 2b) by (Figure 2b)
Line 193. It is necessary to change the (Fig. 2c) by (Figure 2c)
Line 260. …..in Supplementary data (Figure S2). not exist this document with Supplementary data
Line 282. It is suggested to change …in ref. [34], by .....this dependency was theoretically substantiated by what was reported by Rothschild and Komem, (2004) [34].
Line 311. It is suggested to complement ….stationary state (equation 2):
In reaction, it is suggested to put ® and not =
Line 372. It is suggested to complement …. and OH-groups (equation 3):
In reaction, it is suggested to put ® and not =
Line 393. It is suggested to complement ….according to reaction 4
In ecuation 4 , it is suggested to put ® and not =
Although the document contains some interesting elements of discussion, I think it should be included a section of Discuss of results, because in the conclusions section the results of the sensitivity of the sensors to benzene are discussed a little, but not all the characterization results were discussed extensively with the sensibility; if this section were included, it would allow the conclusions paragraph to be reduced, which is very extensive.
Author Response
We are grateful to the Reviewer for the careful detailed review and valueble comments helping us to improve the article. Addressing the suggestions, we hope that we succeeded in improving the presentation of the work. Changes in the revised text are marked by yellow marker.
1) "Line 57 to 72. In this section, the objective(s) of the work are usually stated based on previously presented information so that the scientific contribution of the work is highlighted. It is necessary to improve and to reduce, because is widely described and it is not necessary present results."
Authors reply:
This paragraph was revised and shortened. The aims and novelty of the work were formulated, while presentation of results was reduced.
2) "Line 73 to 80. The information in the table should serve to delve deeper into the justification and relevance of the work, because it is only presented and mentioned that the MOS differ in fundamental parameters. Or, it is suggested to switch to section 2. Materials and Methods, because the properties of the semiconductors to be prepared are detailed in the table."
Authors reply:
We switched the Table with the fundamental properties of materials into section 2. Materials and Methods.
3) "Materials and Methods
Between Line 81 and line 82. It is suggested to put subtitle 2.1. Sensor Preparation
Between Line 98 and line 99. It is suggested to put subtitle 2.2. Sensor Characterization
Between Line 125 and line 126. It is suggested to put subtitle 2.3. Evaluation of Sensitivity of Sensor
Line 107-109. It is suggested to change micrographs for microscopy, because the technique is the microscopy and not micrographs
Line 133. It is necessary to change the “2,0-3,5” by “2.0-3.5”
Line 140. It is suggested to complement ….according to equation 1
Line 141. It is suggested to change … Rg – resistance in test gas, by …where Rg is resistance in test gas.
In this section is necessary to describe the characterization for TPD because because the results are being presented."
Authors reply:
All the comments were addressed and corrections are made in the revised manuscript. TPD experiment description is added in the section.
4) "Results and Discussion
Line 148-149. not exist this document with Supplementary data.
Line 166. It is necessary to change the (Fig. 1e,f) by (Figure 1e,f)
Line 176 and 208. It is necessary to change the (Fig. 2a) by (Figure 2a)
Line 182. It is necessary to change the (Fig. 2b) by (Figure 2b)
Line 193. It is necessary to change the (Fig. 2c) by (Figure 2c)
Line 260. …..in Supplementary data (Figure S2). not exist this document with Supplementary data
Line 282. It is suggested to change …in ref. [34], by .....this dependency was theoretically substantiated by what was reported by Rothschild and Komem, (2004) [34].
Line 311. It is suggested to complement ….stationary state (equation 2):
In reaction, it is suggested to put ® and not =
Line 372. It is suggested to complement …. and OH-groups (equation 3):
In reaction, it is suggested to put ® and not =
Line 393. It is suggested to complement ….according to reaction 4
In ecuation 4 , it is suggested to put ® and not = "
Authors reply:
Supporting information file is now added to the revised manuscript, it was a mistake during the submission process.
All the corrections were made addressing the comments. We are not sure that we correctly interprted the symbol which the Reviewer suggests to put into reactions instead of =. Probably, right arrow was meant? It is reasonable to use arrows in the reactions. We changed the equality signs = for the right arrows.
4) "Although the document contains some interesting elements of discussion, I think it should be included a section of Discuss of results, because in the conclusions section the results of the sensitivity of the sensors to benzene are discussed a little, but not all the characterization results were discussed extensively with the sensibility; if this section were included, it would allow the conclusions paragraph to be reduced, which is very extensive."
Authors reply:
Following the Reviewer recomendation, we subdivided the Results and Discussion section into the section of Results and the Discussion sections, which allowed to make formulate the Conclusions in a concize manner.
Reviewer 3 Report
The authors investigated the benzene-detection difference between pristine and PdOx-modified n-type MOX (ZnO, In2O3, SnO2, TiO2, and WO3) sensors in terms of active sites, surface acidity, and metal-oxygen bond energy via DRIFT, XPS, and TPR methods. The work is well written and the results sound very interesting. A minor revision is suggested with some issues listed below:
1) Recent work about metal oxide based heterojunctions have been employed for gas recognition (such as ACS Appl. Mater. Interfaces, 2021, 13, 47, 56485-56497), and is helpful to enrich the research background. In addition, the authors are suggested to introduce the similar work in the introduction part to further showcase the novelty of this work.
2) The supporting information file is absent.
3) In Fig. 5b, all metal oxide based sensors showed nearly monotonous response increase with operation temperature. As is well known, the typical MOX sensors undergoes a volcano-shaped response evolution. Does it mean that the proposed conclusion in this work was partially tenable over the investigated temperature regime?
Author Response
We thank the Reviewer for such an appreciation of the article. Following the comments, we made some changes in the revised manuscript which are marked by green marker.
1) The cited reference was added into the Introduction section. Also, some up-to-date literature on benzene sensing materials were cited in the Introduction section and in the Results section (Table 3) in the revised manuscript.
2) We thank the Reviewer for notice. It was a mistkae during manuscript submission. The Supplementary file is now added.
3) The Reviewer is absolutely right about the volcano type relations of sensitivity to temperature. In the present work, we could not exceed the operation temperature of PtOx-modified samples above the annealing temperature of nanocomposites, in order not to affect the morphology of PtOx nanoparticles. For the sake of comparison, the prisine MOS were also operated at same temperature, not higher than 220 C. Since this temperature is not high enough, comparing to typical benzene sensors in literature (300-400 C), we did not observe the maximum on the sensitivity-temperature relations. Likely, is due to high enough activation energy of benzene oxidation because of thermodynamic stability of the aromatic ring. Thus, the higher operation temperature is preferable, i.e. at the limit of annealing temperature of MOS/PtOx samples.
Reviewer 4 Report
The authors have analyzed different metal oxides alone and combined with PtOx as thick film sensing materials for benzene. The paper is overall well written with a good and detailed description of the synthesis procedure by chemical aqueous deposition for metal oxides used and their characterization by XRD, TEM, XPS followed by TPR analysis enabling determination of oxidative sites for Pt doped and pure metal oxide samples.
TPD investigation of pure samples enabled determination of the acid cites on pure metal oxide samples, but not on Pt doped, so some additional clarification to the relevance of these measurements needs to be given in view of the relevance of Pt addition.
Concerning the sensitivity analysisto benzene, the authors need to include a detailed analysis of the sensitivity and response to benzene of different metal oxides, both pure and with Pt addition in the form of thick film sensors. The analysis according to BET surface is not clear and justified completely. The authors need to also compare actual thick film sensor performance and compare sensitivity to different benzene concentrations and also the response values (were they different for different sensing materials, in terms Mohm, kohm). Did Pt addition change the measured resistance and how for each of the materials used? How does the sensitivity compare to other current applied sensing materials for benzene?
Justification of the assumption that Pt does not affect acid sites needs to be given as this is only an assumption and the values determined for pure samples were used in Fig. 6b. This needs some more literature justification. Why does Fig. 6b not include the determined H2TPR active sites for Pt doped MOS samples that was measured?
Images of the porous layer surfaces would be an added benefit to understand the influence of specific surface area on the sensing performance. Also an illustration of the sensing device/electrodes and thick film surface would be good.
Author Response
1) "The authors have analyzed different metal oxides alone and combined with PtOx as thick film sensing materials for benzene. The paper is overall well written with a good and detailed description of the synthesis procedure by chemical aqueous deposition for metal oxides used and their characterization by XRD, TEM, XPS followed by TPR analysis enabling determination of oxidative sites for Pt doped and pure metal oxide samples."
Authors reply:
We are grateful to the Reviewer for the high appreciation of our work and appropriate comments that hopefully helped us to improve the manuscript. The changes made in the revised version following the Reviewer comments are marked by cian marker.
2) "TPD investigation of pure samples enabled determination of the acid cites on pure metal oxide samples, but not on Pt doped, so some additional clarification to the relevance of these measurements needs to be given in view of the relevance of Pt addition."
Authors reply:
We agree with the notice. The problem with TPD is the use of increasing temperature to monitor the desorption of ammonia preliminary adsorbed onto surface acid sites. The materials with prominent catalytic oxidation activity (e.g. modified by noble metals) can often not to be adequately tested by TPD since ammonia is also a reducing gas and actually it was oxidized to NOx species prior to desorption, as was observed by mass spectrometry. To support our assumption on the similar surface acidity of pristine MOS and PtOx-modified materials, we have performed additional experiments. The materials were studied by in situ diffuse reflectance infrared spectroscopy (DRIFT) under the conditions of ammonia adsorption at room temperature. Due to the low temperature, ammonia is not oxidized at room temperature and is just adsorbed in the form of NH3 molecules bound to Lewis acid sites (surface cations) or NH4+ ions bound to Broensted acid sites (OH-groups). These new results are added and discussed in the revised manuscript. Briefly, the peaks of NH3 and NH4+ adsorbates were both observed with similar intensities on the spectra of MOS and MOS/PtOx materials. It is an evidence of the insignificant effect of PtOx on the surface acidity, likely becuase of low percentage of the additive (1 wt.%) and therefore sparce coverage of MOS by PtOx nanoparticles.
3) "Concerning the sensitivity analysisto benzene, the authors need to include a detailed analysis of the sensitivity and response to benzene of different metal oxides, both pure and with Pt addition in the form of thick film sensors. The analysis according to BET surface is not clear and justified completely. The authors need to also compare actual thick film sensor performance and compare sensitivity to different benzene concentrations and also the response values (were they different for different sensing materials, in terms Mohm, kohm). Did Pt addition change the measured resistance and how for each of the materials used? How does the sensitivity compare to other current applied sensing materials for benzene?"
Authors reply:
We revised the presentation and discussion of sensing results and showed the absolute resistance responses and sensor signals in the main text and Supplementary data in the revised submission. Briefly, the resistance of all PtOx-modified sensors was higher than that of pristine MOS, because of the electronic cluster-support interaction. Electronic acceptation by supported PtOx from definite MOS was also inferred from XPS data. As a result, the semiconductor oxides had electronic depletion in presence of PtOx and the resistance increased. Also, the concentrational dependences of sensor signals of all samples were added as a Supplementary data and discussed in the main text of the revised manuscript. The comparison of sensors behavior with currently researched benzene sensors inliterature is added as Table 3 and an according discussion in the revised manuscript. Also, the justification of sensor signals normalization by the BET area is revised in the article to make it more detailed. It has often been noticed that experimental sensor signals to reducing gases are inverse proportional to mean particle size, and it was theoretically justified in the work by Rothschild and Komem (J.Appl.Phys. 2004) that sensitivity is directly proportional to surface-to-volume ratio of a sensor material, i.e. the number of surface atoms divided by the number of atoms in the bulk. The surface-to-volume ratio is proportional to specific surface area of a material, with a constant factor including an average surface density of atoms, molar mass and Avogadro number. Thus, since the sensitivity is expected to be proportional to BET area and the experimental samples were not obtained with the same microstructure parameters, we calculated the effective sensor signal. This is a signal expected for a material with a given chemical composition if it had the BET area of 50 m2/g, as an average specific surface area that can in principle be reached for most of the studied oxides via thorough tailoring the synthesis conditions.
4) "Justification of the assumption that Pt does not affect acid sites needs to be given as this is only an assumption and the values determined for pure samples were used in Fig. 6b. This needs some more literature justification. Why does Fig. 6b not include the determined H2TPR active sites for Pt doped MOS samples that was measured?"
Authors reply:
The reply to this comment and changes made in the revised text are essentially the same as those given to the Reviewer comment 2 above. We could not find literature data on the comparative surface acidity of pristine and Pt-modified oxides. Therefore, we perfomed the additional DRIFT study of ammonia adsorption at the surface of MOS and MOS/PtOx materials, which is presented and discussed in the revised manuscript. The concentration of oxidative sites (H2 TPR) is not included in the X-axes in the plot of MOS/PtOx sensitivity in relation to acid sites concentration (Figure 8b of revised manuscript), becuase otherwise (if acid sites and oxidative sites were summed for MOS/PtOx) no adequate correlation could be observed for the sensitivity to benzene. As we mention at the end of discussion in the revised text, the oxidative sites may be not determinig the sensitivity of MOS/PtOx because of the dominant impact of PtOx to the catalytic oxidation of benzene. The catalytic activity in its turn is strongly dependent on surface acidity of supporting MOS. This is essentially the difference in sensing mechanisms by pristine MOS and MOS/PtOx observed by DRIFT spectroscopy. In case of pristine MOS, i.e. at the absence of catalytic PtOx nanoparticles, the oxidative sites at the oxides surfaces play a more improtant role in benzene oxidation, as well as acid sites do.
5) "Images of the porous layer surfaces would be an added benefit to understand the influence of specific surface area on the sensing performance. Also an illustration of the sensing device/electrodes and thick film surface would be good."
Authors reply:
We added the images of sensor substrates before and after the thick film deposition in the Supplementary data of the revised manuscript. We did not have the opportunity to registed the SEM micrographs of all the materials, due to the limited time for revision. However, it looks like the SEM images are not representative to visualize the difference in specific surface areas of materials. In the attached file, there are SEM images that we have got for SnO2 and TiO2. Despite the materials differ in more than 10 times in the BET areas (more than 90 m2/g for SnO2, and about 7-8 m2/g for TiO2), the SEM images of porous layers of both oxides look quite similar. So that, the use of BET area value is more appropriate, since it is actually the surface area available for gases, including micropores which cannot be visualized by SEM.

Round 2
Reviewer 4 Report
The authors have answered all the questions raised in the review well and in detail and they have implemented appropriate changes to their paper. I recommend this work for publication in its present form.